# Silicon Electrodeposition for Microelectronics and Distributed Energy: A Mini-Review

Andrey Suzdaltsev 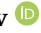

Institute of Hydrogen Energy, Ural Federal University, Mira St. 19, 620075 Yekaterinburg, Russia; a.v.suzdaltsev@urfu.ru

**Abstract:** Due to its prevalence in nature and its particular properties, silicon is one of the most popular materials in various industries. Currently, metallurgical silicon is obtained by carbothermal reduction of quartz, which is then subjected to hydrochlorination and multiple chlorination in order to obtain solar silicon. This mini-review provides a brief analysis of alternative methods for obtaining silicon by electrolysis of molten salts. The review covers factors determining the choice of composition of molten salts, typical silicon precipitates obtained by electrolysis of molten salts, assessment of the possibility of using electrolytic silicon in microelectronics, representative test results for the use of electrolytic silicon in the composition of lithium-ion current sources, and representative test results for the use of electrolytic silicon for solar energy conversion. This paper concludes by noting the tasks that need to be solved for the practical implementation of methods for the electrolytic production of silicon, for the development of new devices and materials for energy distribution and microelectronic application.

**Keywords:** silicon; electrodeposition; molten salts; Si-anode; lithium-ion battery; solar energy





## 1. Introduction

In the context of a global increase in energy consumption and a depletion in energy resources, increasing attention is being paid to the development of new materials and devices to increase the share of renewable energy use [1–3]. In turn, the tasks facing microelectronics include the development of new multilayer and hybrid semiconductor structures, as well as the reduction of costs for the synthesis of semiconductor materials.

Silicon-based materials are especially in demand for the creation of microelectronics and distributed energy devices. In particular, the possibility of using silicon and silicon-based materials in solar energy conversion devices and energy-storage devices is being actively studied [4,5]. Silicon-based materials remain the basis of photoconverters, and the replacement of graphite anodes with silicon can increase the capacity of lithium-ion current sources by an order of magnitude (theoretically, from 372 to 4200 mAh g$^{-1}$ [4,6]). The efficiency of such devices can be ensured by using high-purity micro-sized silicon films with a controlled content of micro-impurities (photoelements) or nano-sized and submicron silicon particles (lithium-ion current sources). Nano-sized clusters of high-purity silicon with a controlled content of micro-impurities are in demand in microelectronics [7,8]. Silicon is also widely used in metallurgy (steel deoxidation, synthesis of alloys) and organosilicon chemistry (oils, silicones, etc.), for the manufacture of laser devices, and for the production of hydrogen (ferrosilicon) [9]. Silicides of various metals can also play an important role [10].

Currently, metallurgical silicon is obtained by carbothermal reduction of quartz at a temperature of about 1800 °C [11], while the production of high-purity silicon is based on the Siemens process [12]. The Siemens process is characterized by its multistage nature, relatively high energy and material costs, and relative complexity of execution. However, there are no alternative technologies yet available for industrial pilot implementation.

Since the 1970s, methods have been actively developed to obtain high-purity silicon, including the electrodeposition of silicon from molten salts. These methods are relatively simple and cheap, since they make it possible to obtain silicon in one to three stages [13–16]. Furthermore, silicon can be obtained directly from quartz. Moreover, the purity and morphology of obtained silicon allows its use in photoconverters (thin films) and metal-ion current sources (nano- and micro-sized fibers, needles, tubes) without its additional recrystallization [17]. The need for additional purification of electrolytic silicon by recrystallization methods in order to achieve semiconductor purity remains questionable. Recently, methods have been proposed for obtaining silicon and silicon-based materials by electrodeposition from ionic liquids and organic electrolytes [18,19]. These methods are of interest, although their industrial implementation will require large volumes of expensive reagents.

It should also be noted that silicon can be obtained by electrolysis of electrolytes without hydrogen. Furthermore, silica films and inorganic silicon-containing clusters can be obtained by electrolysis of aqueous solutions [20,21]. Electrolysis with a milder potential allows "electro-click" hybrid films [22], and electropolymerization allows developers to obtain organic films [23].

Figure 1 shows schematic diagrams of the implementation of methods for obtaining silicon via the Siemens process and by electrodeposition from molten salts. In both cases, silicon can be obtained from quartz, while the Siemens process includes several energy- and material-intensive operations. From the illustrated scheme, it can be seen that the electrolytic production of silicon can be carried out in one to three stages, and the application range of silicon is broad, including thin films, ultra-sized fibers, and nuclei, which, after separation of electrolyte residues, can be used in electrochemical devices. In the Siemens process, relatively large micro-sized dendrites of polycrystalline silicon are obtained, which cannot be directly used in energy conversion or storage devices.

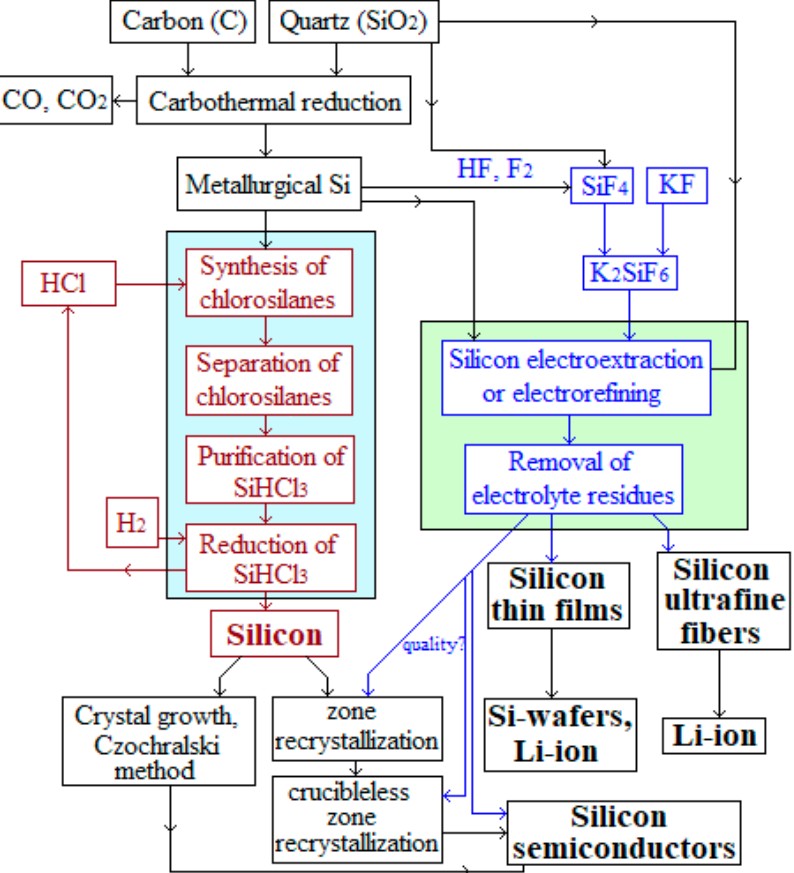

**Figure 1.** Schemes for obtaining silicon.

Both methods (Siemens process and electrolysis) provide polycrystalline silicon, which can be recrystallized into a monocrystalline phase. Meanwhile, epitaxial deposits on different substrates can be obtained by electrolysis. Obtaining monocrystalline silicon is characterized by higher cost and a large volume of recycled silicon, but this silicon is slightly superior to polycrystalline silicon in terms of the efficiency of solar energy conversation. Polycrystalline silicon is cheaper, and the imperfection of its macrostructure does not prevent it from being utilized to convert solar energy. There are currently many works aimed at sensibilization of the silicon surface [24].

In the present review, a brief comparative analysis of modern methods of silicon electrodeposition is presented, and the results and prospects for the use of electrodeposited silicon in semiconductor devices, energy conversion, and storage devices are analyzed. As can be seen from the diagram in Figure 2 [25], the most popular applications of silicon remain solar energy and semiconductor materials, while the development of lithium-ion current sources with Si-based anodes has only recently gained popularity.

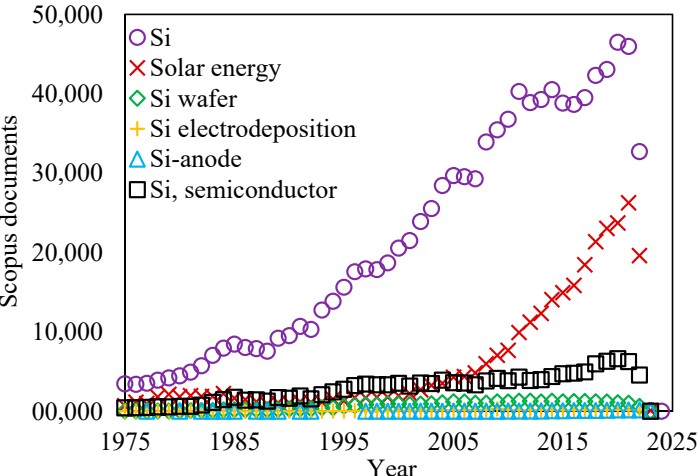

**Figure 2.** Analysis of sources of scientific information on different areas of silicon use [25].

## 2. Molten Salts for Silicon Electrodeposition

### 2.1. Choice of the Molten Salt

The efficiency of silicon production by electrolysis of molten salts can be achieved with an optimal combination of electrolyte composition and electrolysis parameters. In the general case, the production of silicon by electrolysis of molten salts involves several operations, including purification of the initial reagents from impurities, electrodeposition, and separation of the silicon deposits from salt residues. The parameters of these operations affect the composition and morphology of silicon deposits, as well as the efficiency of the process as a whole. In this regard, the following factors should be taken into account when choosing molten salt and electrolysis parameters:

- purity and low chemical activity of salts in relation to the materials of the electrolyzer, and the possibility of their purification;
- stability of the concentration and composition of silicon-containing electroactive ions, which can be ensured by the high complexing ability of silicon ions;
- stability of elemental silicon in melts with $Si^{4+}$ ions;
- rate of silicon electrodeposition, provided by a consistently high concentration of electroactive silicon ions, in consideration of the laws of their electroreduction;
- high solubility of salts in aqueous solutions or high vapor pressure of salts during high-temperature distillation.

Simultaneous observance of all these factors is practically impossible, and as a result the effectiveness of the use of certain compositions of molten salts must be tested empirically.

## 2.2. Basics of the Silicon Electrodeposition

In the available works, attention was mainly focused only on the study of the kinetics of the electroreduction of silicon-containing electroactive ions, as well as on determining the parameters of silicon electrodeposition with an expected morphology. Silicon electrodeposition includes the electroreduction of silicon-containing ions at the cathode in one or more electrode steps, depending on the melt composition and electrolysis parameters according to the overall reaction (1):

$$Si^{4+} + 4e = Si^{0} \tag{1}$$

The presence of a side reaction of disproportionation in melts has been reported (2):

$$Si^{0} + Si^{4+} = 2Si^{2+} \tag{2}$$

Reaction (2) leads to a decrease in the cathodic current efficiency and a change in the kinetic parameters of silicon electrodeposition. To date, the regularities of silicon electrodeposition have been well studied, and the fundamental possibility of production of silicon with controlled morphology has been reported by varying such parameters as current density, cathode potential, melt composition, and electrolysis mode (pulse, reverse, etc.). Despite the positive results, that work has not been brought to practical implementation. Comparatively neglected issues include the cathodic current efficiency of silicon, the influence of the semiconductor nature of silicon, silicon purity, and methods of post-treatment.

## 2.3. Results of the Silicon Electrodeposition

To date, the greatest attention has been paid to the targeted production of silicon for energy conversion and storage devices, mainly from molten $CaCl_2$–$(NaCl)$–$CaO$–$SiO_2$ ($CaSiO_3$) [26–28] and $KF$–$KCl$–$K_2SiF_6$ [29–31] with operating temperatures of 800–860 and 700–750 °C, respectively (see Table 1). The disadvantages of chloride–oxide melt are its relatively high temperature, low rates of silicon electrodeposition, and the presence of oxides in the melt. The oxides are inevitably included in the pores of the deposit, and probably degrade the performance characteristics of silicon when used in semiconductor devices, energy conversion, or storage devices. In turn, the disadvantage of the fluoride–chloride system is its relatively high chemical activity, which leads to corrosion of the structural materials of the reactor and complicates the production of high-purity silicon. Despite disadvantages, researchers [26–31] have reported silicon deposits obtained by electrolysis of $CaCl_2$–$CaO$ and $KF$–$KCl$-based melts in the form of fibers (from 30 to 500 nm), dendrites, thin films, and other morphologies. The declared purity of electrolytically obtained silicon reaches 99.99 wt% or more if the impurities of the electrolyte components are not taken into account [28].

We carried out a series of experiments on silicon electrodeposition from low-fluoride systems based on mixtures of KCl, CsCl, and LiCl with additions of $K_2SiF_6$ and $SiO_2$ in the temperature range 350 to 790 °C [32–36]. Due to the possibility of deep purification of chlorides by zone recrystallization [37], the proposed systems are suitable for use to obtain high-purity silicon. The disadvantage of low-fluoride systems is the lower complexing capacity of silicon, which can be improved by increasing the proportion of CsCl in the melt. As a result, we also obtained silicon deposits in the form of thin (1–5 μm) films, as well as submicron (50 to 300 nm diameter) fibers, filaments, and tubes.

**Table 1.** Parameters and results of silicon electrodeposition from molten salts.

| Electrolyte [Refs] | Si Source | $T$ (°C), Current Density (A cm$^{-2}$) | Results | Images |
|---|---|---|---|---|
| Mixtures of KF, NaF, LiF, BaF$_2$, CaF$_2$ [13–15] | Si, K$_2$SiF$_6$, SiO$_2$ | 550–1500, 0.05–1 | Compact deposits up to 1 mm, micro-sized dendrites, fibers |  |
| CaCl$_2$–CaO [26–28] | SiO$_2$, CaSiO$_3$ | 800–860, 0.01–0.05 | Micro- and nano-sized fibers, films on quartz, tubes |  |
| KF–KCl [29–31] | Si, K$_2$SiF$_6$, SiO$_2$, SiCl$_4$ | 700–750, 0.05–0.2 | Micro- and nano-sized fibers, films, dendrites |  |
| Mixtures of KCl, CsCl, LiCl with K$_2$SiF$_6$ [32–36] | Si, K$_2$SiF$_6$, SiO$_2$ | 350–790, 0.05–0.4 | Micro- and nano-sized fibers, needles, pipes, films |  |
| KI–KF–KCl [38,39] | Si, K$_2$SiF$_6$ | 700–750, 0.05–0.2 | Micro- and nano-sized fibers, films |  |
| Ionic liquids, organic electrolytes [18,19,40] | SiCl$_4$, chloro-silanes | 25–80, 0.0001–0.005 | Micro- and nano-sized fibers, films |  |

The continuous appearance of new works devoted to the development of methods for producing silicon and silicon-based materials indicates the presence of shortcomings in the existing methods and the relevance of searching for new energy-efficient and resource-saving methods for producing silicon. In particular, these concerns relate to works

aimed at the synthesis of silicon from iodide melts [38,39], organic electrolytes, and ionic liquids [16,17,40].

Table 1 shows the parameters and typical results of silicon electrodeposition from molten salts. As noted in Figure 1, ultrafine silicon fibers are of interest for the development of lithium-ion current sources, while thin silicon films are required for solar energy conversion and for current sources. The most representative results of the use of electrodeposited silicon from these melts are summarized in the following sections. Table 1 shows only the results of studying the obtained silicon by SEM (sizes, morphology), although XRD (lattice parameters) and ICP MS (elemental composition) methods can also be used for this purpose. Less commonly used methods are TEM (sizes), XPS (energy characteristics of Si-Si bonds), Raman spectroscopy (characteristics of Si-Si bonds), nuclear microanalysis (micro-impurities, isotopes), etc.

### 3. Electrolytic Silicon for Microelectronics

The first proposals for obtaining silicon by electrolytic methods were presented in terms of its use in semiconductor materials and microelectronics. However, at present there are no targeted works describing a full cycle of research on the production of silicon and its application in semiconductor materials, although a number of works have made statements about the possibility of electrodeposition of silicon of the *n*-, *p*-, or mixed *p-n*-type [26]. In the current author's opinion, the absence of such works is caused by the complexity of the experimental choice for the operation to attain additional purification of silicon from impurities and electrolyte residues.

### 4. Electrolytic Silicon for Lithium-Ion Current Sources

The operability of a lithium-ion current source with silicon-based anodes can be ensured by using silicon with a developed surface, as well as by thin silicon films [41].

During electrolysis of a $CaCl_2$–$CaO$–$SiO_2$ melt, nanosized fibers, particles, wires, and tubes were obtained on a nickel cathode, depending on the electrolysis potential, at a cathode current density of 80–100 mA cm$^{-2}$ [4]. The obtained silicon tubes had the highest specific surface area (99.9 m$^2$ g$^{-1}$) and the best lithiation–delithiation characteristics (specific capacity after 1000 cycles: 3044 mAh g$^{-1}$ at a current of 0.2 A g$^{-1}$, and 1033 mAh g$^{-1}$ at 1 A g$^{-1}$). Other studies have also reported the production of nanosized silicon deposits with discharge capacity from 500 to 3500 mAh g$^{-1}$, depending on their morphology and purity.

The results of tests involving a lithium-ion current source of silicon electrodeposited from KF–KCl–$K_2SiF_6$-based melts are extremely limited. In particular, nanosized silicon particles (25–50 nm) and fibers (diameter 150–250 nm, length 1–4 μm) with a specific surface area of 14–15 m$^2$ g$^{-1}$ were obtained [31] at a temperature of 700°C and a cathode current density of 10–20 mA cm$^{-2}$. The principal possibility of lithiation or delithiation of the obtained silicon was demonstrated.

As a result of the electrolysis of KCl–$K_2SiF_6$, KCl–$K_2SiF_6$–$SiO_2$, KCl–CsCl–$K_2SiF_6$, and LiCl–KCl–CsCl–$K_2SiF_6$ melts, we obtained silicon deposits of various morphologies by varying the electrolysis parameters [32–36]. In particular, at a cathode current density of 20 to 150 mA cm$^{-2}$, silicon fibers (diameter 100–700 nm), tubes, and needles (diameter 100–400 nm) were obtained, the specific capacity of which after 15 cycles varied from 200 to 850 mAh g$^{-1}$.

Along with pure silicon, Si/C mixtures and composites [42,43], which can also be obtained by electrolysis of molten salts, are considered promising anode materials for lithium-ion current sources.

In general, a significant improvement in the energy characteristics of lithium-ion current sources can be obtained by replacing graphite anodes with silicon-basic ones. However, certain important technical issues remain unresolved, and it is necessary to address these in order for such anodes to have practical implementation.

## 5. Electrolytic Silicon in Solar Cells

The most demanded photoconverters are continuous silicon films with a thickness of 10–20 μm and a given content of donor micro-impurities.

In [5], the effects were studied of substrate material (Ag, Mo, C), electrodeposition potential, and particle size distribution of $SiO_2$ on the morphology of silicon deposits obtained by electrolysis of a $CaCl_2$–$CaO$–$SiO_2$ melt at a temperature of 855 °C. A continuous photosensitive silicon film 180 μm thick was obtained on graphite at the lowest cathodic overvoltage. The authors noted the need for periodic purification of the melt from unwanted impurities, by purification electrolysis.

In [44], under conditions of potentiostatic electrolysis of a $CaCl_2$–$CaO$–$SiO_2$ melt at 850 °C, silicon films 20–25 μm thick were obtained with *p*-, *n*-, and mixed *p-n*-conductivity. The photosensitivity of the obtained silicon films was demonstrated as being 3.1% more effective than commercial analogues. The same authors obtained silicon films from 10 to 60 μm thick, with n-conductivity, by electrolysis of a $KCl$–$KF$–$K_2SiF_6$ melt on graphite at a temperature of 650 °C. To increase the number of electrocrystallization centers, 0.020–0.035 wt% tin was added to the melt [45]. In the opinion of the authors, the presence of up to 0.35 wt% tin in the obtained silicon films should not affect their photosensitivity, which was up to 55% of commercial samples.

In [46], the effects of cathode current density, substrate material, source ($K_2SiF_6$, $SiCl_4$), and silicon ion concentration in the $KF$–$KCl$ melt on the morphology of electrolytic silicon deposits were studied for a temperature of 750 °C. The optimal conditions for obtaining smoothed silicon films with a thickness of 20 to 60 μm were determined and their photosensitivity were demonstrated.

Several works have noted the possibility of obtaining continuous silicon films with purity of 99.9–99.99 wt%, doped with impurities such as B, Al, etc. We also carried out preliminary studies that showed the possibility of electrodeposition of photosensitive silicon films with a thickness of 1 μm, during electrolysis of $LiCl$–$KCl$–$CsCl$–$K_2SiF_6$ melts [36].

## 6. Conclusions

It follows from the above analysis that silicon electrodeposition is primarily of interest for the creation of new energy-conversion and storage devices with improved performance. Less attention has been paid to the electrodeposition of silicon to meet the requirements of microelectronics.

The most actively studied methods of silicon electrodeposition include the electrolysis of $CaCl_2$–$(NaCl)$–$CaO$–$SiO_2$ ($CaSiO_3$) and $KF$–$KCl$–$K_2SiF_6$ melts with operating temperatures of 800–860 and 700–750 °C. Silicon deposits of various sizes and morphologies have been obtained, the possibility of doping silicon with micro impurities for use in energy conversion and storage devices has been shown. Alongside this, an active search is underway for new methods for the electrodeposition of silicon and silicon-based materials from molten salts, ionic liquids, and organic electrolytes.

For the practical implementation of the developed methods of silicon electrodeposition, as well as for the creation of new materials and devices for energy distribution and microelectronics, it is necessary to address more actively the issues related to the purification of electrodeposited silicon from electrolyte residues, and directly to consider the design of silicon-based materials and devices.

**Funding:** This work was carried out in the frame of the State Assignment number 075-03-2022-011 dated 14.01.2022 (the theme number FEUZ-2020-0037).

**Institutional Review Board Statement:** Not applicable.

**Informed Consent Statement:** Not applicable.

**Data Availability Statement:** Not applicable.

**Conflicts of Interest:** The authors declare no conflict of interest.

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
