# Peer review of "Silicon Electrodeposition for Microelectronics and Distributed Energy: A Mini-Review"

_2673-3293, doi:10.3390/electrochem3040050_

Round 1

Reviewer 1 Report

The present perspective by Suzdaltsev revisits the existing electrocatalytic approaches for the production of silicon to use in photovoltaics and energy storage devices. The main technological challenges are highlighted to direct future research in sustainable production. The review focuses on a very important topic and will gain the wide interest of the scientific community. Though the article provides a glimpse of the current state of the art, the in-depth scientific discussion is missing, especially for novices.

1.  The quality of text can be improved by adding some cost analysis of the conventional carbothermal process, and a comparison of polycrystalline and monocrystalline silicon and their efficiency.

2.  Methods used for the characterization of electrochemically produced nanostructure should be discussed. Which parameters govern the morphology and crystal facet during electrodeposition?

3.  Is there any understanding of the electrodeposition process, mobility of ions and challenges faced at high-temperature electrodeposition?

4.  A tentative schematic showing the deposition process will be helpful.

Overall, It is recommended to improve the readability of the manuscript by adding more scientific and practicality of the process. Also, alternate applications beyond these two can be discussed in brief. 

Author Response

Please, find attached file.

Reviewer 2 Report

This Mini-review from Andrey Suzdaltsev introduces recent past and recent progresses in the field of Silicon electrodeposition. The structure of the review is great and the topic (althougth very specific) is well treated, including challenges, limitations and application outlooks . I have therefore very few comments on this work:

- On the form of the manuscript: there are too many sub-paragraphs in the text that can simply be merged.

- Table 1 greatly demonstrates the variety of morphologies that can arise from Silicon electrodeposition. However, there is no detailled explanation on the added value of such well defined morphologies (if any). Adding a few lines to link the obtained structures with target properties would complement the review. 

- Electrodeposition as a technic is poorly introduced in the paper. Adding a few lines to explain the audience what is electrodeposition and what it enables in broader fields is needed (e.g:  water electrolysis allows the deposition of silica films (Nature Materials volume 6, pages 602–608 (2007) and inorganic clusters (J. Mater. Chem. C, 2017, 5, 10477-10484) milder potential allow "electro-click" hybrid films (Phys. Chem. Chem. Phys., 2018, 20, 2761-2770) and of course electropolymerization of organic films (Nanomaterials 2019, 9(8), 1125))

Author Response

Please, find attached file.

Reviewer 3 Report

1. The abstract section should be written in paragraph form instead of point by point.

2. The authors wrote: "In the context of a global increase in energy consumption and a reduction in energy 25 resources, increasing attention is being paid to the development of new materials and 26 devices to increase the share of renewable energy use.....Herein, the authors are suggested to add some references related to energy issues. Please see (ACS Applied Energy Materials 4 (12), 14043-14058, Journal of Materials Chemistry A 9 (20), 12019-12028, Sustainable Energy & Fuels 5 (11), 2960-2971).

3. The authors wrote: "One of the materials in demand for the creation of microelectronics and distributed 30 energy devices are silicon-based materials. In particular, the possibility of using silicon- 31 based materials in solar energy conversion devices and energy storage devices is being 32 actively studied". Please add some pros of silicon.

4. There are several typing mistakes and text is little complicated, which should be simplified.

Author Response

Please, find attached file.
